# Characterization of Virulence Factors and Antimicrobial Susceptibility of *Streptococcus agalactiae* Associated with Bovine Mastitis Cases in Thailand

**DOI:** 10.3390/ani14030447

**Published:** 2024-01-30

**Authors:** Sirirat Wataradee, Thanasak Boonserm, Sukuma Samngamnim, Kittisak Ajariyakhajorn

**Affiliations:** Department of Veterinary Medicine, Faculty of Veterinary Science, Chulalongkorn University, Bangkok 10330, Thailand; 6271008831@student.chula.ac.th (S.W.); thanasak.b@chula.ac.th (T.B.); sukuma.s@chula.ac.th (S.S.)

**Keywords:** antimicrobial susceptibility, dairy cattle, mastitis, multidrug resistance, *Streptococcus agalactiae*, Thailand, virulence factors

## Abstract

**Simple Summary:**

We characterized ten virulent genes from *Streptococcus agalactiae* strains isolated from bovine mastitis (*n* = 100). The strains have at least three virulence genes essential for establishing intramammary infection. The presence of such virulence factors did not show a significant association with clinical and subclinical mastitis. The antimicrobial susceptibility of these strains suggests that beta-lactam antibiotics are recommended for the treatment of *S. agalactiae* mastitis in our region. Interestingly, we identified multidrug-resistant *S. agalactiae* strains (*n* = 7), with two strains being resistant to vancomycin. This evidence highlights significant concerns regarding the judicious use of antimicrobials in mastitis treatment protocols.

**Abstract:**

*Streptococcus agalactiae* is a contagious pathogen that causes bovine mastitis. The ability of *S. agalactiae* to cause widespread mastitis relies on bacterial virulence factors. In this study, we detected 10 virulence determinants associated with mastitis pathogenicity using conventional PCR. The antimicrobial susceptibility of 100 *S. agalactiae* isolates from 13 Thai dairy herds was assessed using the Kirby–Bauer disk diffusion susceptibility test. All strains had at least three virulence factors responsible for invasion, adhesion, and infection (*fbsB*, *bibA*, and *cfb*, respectively). The predominant virulent profile of *S. agalactiae* strains revealed the presence of *fbsA*, *fbsB*, *bibA*, *cfb*, and *cyl* (*n* = 96). Most strains were sensitive to penicillin, ampicillin, amoxicillin-clavulanic acid, cefotaxime, ceftiofur, erythromycin, sulfamethoxazole-trimethoprim, and vancomycin. However, all strains were resistant to aminoglycosides, including kanamycin and gentamicin attributed to the unnecessary antimicrobial use. Furthermore, we identified seven multidrug resistant (MDR) *S. agalactiae* strains among four dairy herds, of which, two were vancomycin resistant. Our study provides profiles for virulence factors and antimicrobial susceptibility, which are beneficial for the clinical monitoring, prevention, and control of bovine mastitis in dairy cattle in Thailand. Moreover, we emphasize the need for awareness regarding the judicious use of antimicrobials on dairy farms.

## 1. Introduction

*Streptococcus agalactiae* is a contagious pathogen associated with bovine mastitis. This is a leading cause of intramammary infection causing economic loss in dairy farms globally. It is an etiological agent of clinical and subclinical mastitis that reduces milk yield and affects milk quality [1,2]. The incidence of bovine mastitis caused by *S. agalactiae* has decreased due to intensive control programs in developed countries [3]. However, China and Denmark have reported it as a reemerging pathogen [2,4]. In Thailand, animal-level prevalence of *S. agalactiae* causing mastitis ranges from 20 to 46.30% [5]. However, in the Khon Kaen province, bulk tank milk analysis indicated that 21.8% of the studied herds were positive for this pathogen [6]. Furthermore, a study indicated that *S. agalactiae* could persist on dairy farms for 2–12 months [7]. The presence of *S. agalactiae* in the herd negatively influenced the total bacterial counts and bulk milk somatic cell counts, demonstrating that the presence of *S. agalactiae* negatively influence milk quality [8].

Capsular genotyping has been extensively applied in epidemiological research into *S. agalactiae* due to the structural diversity of CPS being linked to the genetic multiplicity of the capsular polysaccharide biosynthesis group across various geographic regions [9,10,11,12,13]. The severity of mastitis is primarily determined by the SCC and virulence factors [9]. For example, CPS enables the bacteria to evade the host’s immune defenses. Our previous study reported that the *S. agalactiae* involved in bovine mastitis in Thailand is CPS type Ia [7]. In Asia, CPS type Ia has been consistently reported as the most common type in Thailand and China [14,15,16]. For instance, a study in Thailand on *S. agalactiae* variants causing bovine mastitis revealed that 17 isolates were identified as a single CPS type Ia strain with ST 103 [14]. Similarly, in China, among 102 *S. agalactiae* isolates from subclinical mastitis cases, only two CPS types (Ia and II) were identified, with CPS type Ia being the predominant variant (89%) [16]. The CPS type of *S. agalactiae* (Group B Streptococcus or GBS) is closely associated with its virulence and pathogenicity. CPS is an important virulence factor that plays a multifaceted role in the bacterium’s ability to cause the invasion, adhesion, and immune evasion of the disease [17]. Furthermore, CPS types are associated with virulence levels. This is the target for vaccine development due to its ability to elicit an immune response against the most prevalent and virulent strains in the target area.

The ability of *S. agalactiae* to cause bovine mastitis, encompassing the stages of adhesion, invasion, and infection, relies on bacterial virulence factors. Bacterial virulence factors establish infection. Host immunity is responsible for inflammation due to host-pathogen interaction [18]. Bacteria invade the teat canal and cistern during the invasion stage. *S. agalactiae* colonizes and proliferates in the mammary gland ducts. Virulence factors associated with adhesion and colonization include the immunogenic bacterial adhesion (*bibA*) and protein C, encompassing surface proteins Cα (bca) and Cβ (bac). Virulence factors contributing to invasion and colonization include fibrinogen-binding protein A (*fbsA*), fibrinogen-binding protein B (*fbsB*), pilus island 1 (*PI-1*), and pilus island 2 (*PI-2a*, *PI-2b*). The virulence factor associated with the infection stage is the CAMP factor [11,19]. The CAMP factor (*cfb*) and β-hemolysin/cytolysin (*cyl*) are the primary virulence factors affecting the pathogenicity of *S. agalactiae* [12,17,19]. A CAMP factor is an extracellular protein that induces pore formation in target cells, demonstrated in both in vitro and in vivo systems [20]. The *cyl* gene, which encodes β-hemolysin, causes tissue damage and the systemic spread of the bacteria [21]. These virulence factors are often used as markers for genotyping and epidemiological studies [11,15,17,22,23]. Furthermore, characterizing virulence factors is crucial for understanding clinically relevant virulence factors.

An understanding of the role of virulence factors in bacterial pathogenicity is important for developing targeted treatments for bovine mastitis. For example, virulence factors, such as adhesins and toxins, are crucial to the bacteria’s ability to adhere to and invade mammary epithelial cells, leading to mastitis development. Consequently, treatment approaches may involve targeting these specific virulence factors to disrupt the pathogenic process.

The treatment goals are to cure the infection and eradicate the source of new infections in dairy herds. Intramammary antimicrobial agents effectively eradicated this contagious pathogen [1]. The current standard approach for treating bovine mastitis caused by *S. agalactiae* in dairy cows involves the use of antimicrobial agents. Administering intramammary beta-lactam antimicrobials has been proven to be effective in achieving a high rate of recovery [24,25]. The antimicrobial susceptibility profile of *S. agalactiae* requires regular monitoring. Consequently, it is imperative to conduct antimicrobial susceptibility studies that can facilitate the implementation of rational and appropriate antimicrobial therapy guidelines. This information plays a key role in mitigating the risk of development and spread of antimicrobial resistance (AMR).

AMR in animals has the potential to be transmitted to humans, and this phenomenon is increasingly concerning in the context of public health. This concern underscores the urgent need for comprehensive measures. Individuals working with animals, such as dairy farmers, veterinarians, and animal handlers, may be exposed to AMR-carrying bacteria through direct contact with infected animals [26]. Furthermore, resistant bacteria can persist in the environment [27]. Therefore, regular monitoring and surveillance of AMR in animals are crucial for the measures implemented in the Thai dairy sector.

The aim of this study was to investigate the 10 virulence factors associated with and the antimicrobial susceptibility of *S. agalactiae* strains from bovine mastitis cases in Thailand. We conducted susceptibility tests for 12 antimicrobial agents commonly used to treat bovine mastitis in dairy cows and other infectious diseases in Thailand. The findings of this study have considerable significance as they offer insights into the field of prevention and control strategies, which can enhance the management of bovine mastitis and promote the responsible use of antimicrobial agents in Thai dairy farms.

## 2. Materials and Methods

### 2.1. The Ethical Approval Statement

The protocols employed in this research effort underwent thorough review and received endorsement from the Institutional Biosafety Use Protocol (IBC 1931051) at Chulalongkorn University, Faculty of Veterinary Science.

### 2.2. Bacterial Isolates and Sample Collection

All, 100 strains of *S. agalactiae* examined in this study were identified as CPS type Ia, as documented in a preceding investigation [7]. All studied isolates were identified by conventional microbiological methods by the National Mastitis Council (NMC), the VITEK2 system, and a molecular specific *dlt S* gene. The strains were carefully preserved at −80 °C to facilitate subsequent analyses. The research procedures were carried out in a biosafety level 2 laboratory located at the Livestock Hospital, Faculty of Veterinary Science, Chulalongkorn University, situated in Nakorn Pathom Province, Thailand. From 2016 to 2019, a total of 100 strains of *S. agalactiae* were isolated from mastitis cases in 58 cows, encompassing those diagnosed with clinical mastitis (*n* = 21) and subclinical mastitis (*n* = 37). Clinical mastitis cases underwent assessment by veterinarians through a comprehensive visual examination, including the observation of milk and udder abnormalities (heat, redness, and swelling). Subclinical mastitis was identified through the application of the California Mastitis Test, yielding positive outcomes. Quarter milk culture analysis revealed the presence of *S. agalactiae* in one quarter (*n* = 33), two quarters (*n* = 11), three quarters (*n* = 11), and all quarters (*n* = 3) of the cows (refer to Table 1). These mastitis cases occurred among 13 dairy farms situated in the top three most intensive dairy farming provinces including Nakhon Ratchasima (14°58′14.38″ N, 102°06′7.06″ E), Saraburi (14°31′25.79″ N, 100°54′24.59″ E), and Ratchaburi (13°32′12.16″ N, 99°49′1.63″ E).

The retrieval of strains was conducted from a −80 °C freezer stock in which brain–heart infusion broth with 25% glycerol (Difco™, BBL™ Bacto™) was used as a medium. We employed the culture-based method for recovering studied strains. This technique entails two successive subcultures on sheep blood agar before subsequent testing. In brief, the samples preserved at −80 °C were equilibrated to room temperature. Following this, the samples were plated on blood agar plates using a sterile inoculation loop via the spread plate method. The resultant plates were then incubated in a carbon-dioxide-rich environment at 37 °C overnight. After the incubation period, a singular pure colony was discerned on the plate. Subsequently, it underwent a second round of subculturing, and a sole pure colony was chosen for subsequent detailed analysis.

### 2.3. Reference Strains

The reference bacterial strains, for the laboratory’s quality control measures, were procured from the American Type Culture Collection (ATCC) through LGC Standards. *Staphylococcus aureus* ATCC 25923 served both as a control for the CAMP test and as the positive control for the catalase test. Additionally, various strains of *S. agalactiae* from ATCC, namely ATCC 12400 (CPS type Ia), ATCC 13813 (CPS type II), and ATCC 31475 (CPS type III), were incorporated. All *S. agalactiae* reference strains were included specifically for the identification of virulence genes.

### 2.4. Preparation of the DNA Model for Molecular Procedures

The process of DNA extraction was conducted within a level 2 biosafety cabinet. Subsequently, DNA was extracted from the pure culture in accordance with the DNeasy^®^ Blood & Tissue Kit protocol, as stipulated by Qiagen in Hilden, Germany. The concentration and purity of the extracted DNA were evaluated utilizing a spectrophotometer, specifically the Nanodrop™ lite from Thermo Scientific™ in the Massachusetts, U.S.. The extracted DNA, judged suitable, served as the template for all PCR assays. Subsequently, all DNA samples were carefully collected and stored at −20 °C in preparation for subsequent analyses.

### 2.5. Characterization of Molecular Virulence Factors

Ten virulence genes were determined via singleplex conventional PCR. The virulence factors were *bibA*, *bca*, *bac*, *fbsA*, *fbsB*, *PI-1*, *PI-2a*, *PI-2b*, *cfb*, and *cyl.* These target genes are associated with *S. agalactiae* infection. All primer sequences from this study are displayed in Table 2 [14,26,27]. The PCR mix composition comprises 25 µL, incorporating 12.5 µL of 10x Green GoTaq^®^ Flexi Buffer sourced from PROMEGA in Madison, WI, USA. Additionally, the mixture includes 1 µL [10.0 µM] of the forward primer, 1 µL [10.0 µM] of the reverse primer, 2 µL of the DNA template, and 8.5 µL of Nuclease Free Water. PCR mixtures were amplified using a T100 Thermal Cycler (Biorad^®^, California, U.S.) starting with a 5 min initial denaturation step at 95 °C, followed by 35 cycles of 1 min at 95 °C, 1 min at 55 °C, and 2 min at 72 °C, concluding with a final extension cycle of 5 min at 72 °C.

### 2.6. Antimicrobial Susceptibility of S. agalactiae Strains

The antimicrobial susceptibility test was conducted using the agar Kirby–Bauer disk diffusion susceptibility method following the guidelines of the VET 01 supplement [31] from the Clinical and Laboratory Standards Institute. The CLSI VET01 outlines performance standards for antimicrobial disks and dilution susceptibility tests for bacteria isolated from animals. Twelve antimicrobials (7 classes) were selected for their use in treating bovine diseases in Thailand.

The antimicrobial disks utilized in this study included ampicillin (10 µg), amoxicillin + clavulanic acid (30 µg), cefotaxime (30 µg), ceftiofur (30 µg), gentamicin (10 µg), kanamycin (30 µg), tetracycline (30 µg), erythromycin (15 µg), enrofloxacin (5 µg), penicillin (10 µg), sulfamethoxazole-trimethoprim (25 µg), and vancomycin (30 µg), which were obtained as commercial Oxoid™ disks. The *Streptococcus pneumoniae* ATCC 49619 strain is used as a reference control strain.

The diameters of the zones among the isolates were measured and recorded. The diameters of the inhibition zones were classified as susceptible, resistant, and intermediate according to the breakpoint diameter standard. We applied breakpoints from other animals or other *Streptococcus* species (CLSI, 2019). The susceptibility percentage to an individual antimicrobial was calculated. If the isolate was resistant to three or more antimicrobial classes, the strain was defined as multidrug-resistant (MDR) [32].

### 2.7. Statistical Analysis

The association between the types of mastitis (clinical and subclinical) and the presence of virulence gene categories was determined using the chi-square test. A *p* value of <0.05 was considered statistically significant.

## 3. Results

All *S. agalactiae* were previously identified using the conventional microbiological method by NMC, with a VITEK2 system, and a *dlt S* gene-specific molecular test [7]. All strains of *S*. *agalactiae* collected from 2016 and 2019 were identified based on colony morphology, hemolysis type, esculin hydrolysis, catalase production, and CAMP reaction. To characterize the strains, their phenotypic traits were assessed, encompassing colony growth evaluation and β-hemolysis observation on blood agar (*n* = 88). The remaining 12 isolates were tested for alpha-hemolysis (Figure 1). Furthermore, all tested strains yielded positive results in the CAMP test, while the esculin hydrolysis and catalase tests were negative. These findings collectively enhance the comprehensive understanding of the phenotypic properties and virulence factor profiles of β-hemolysin (*cyl*).

### 3.1. Molecular Characterization of Virulence Factors

We examined 10 virulence factors. The *bibA*, *cfb*, and *fbsB* genes were present in all the strains. The prevalence rates of *fbsA*, *cyl*, and PI-2B were 98%, 97%, and 53%, respectively. However, the *bca*, *bac*, *PI-1*, and *PI-2a* genes were not detected in this study (Table 3). According to hemolytic activity, the *cyl* β-hemolysin/cytolysin gene was identified in 97% of the strains studied, while 88% of the strains exhibited β-hemolysis on blood agar.

The presence of virulence factors was classified into five virulence profiles. It is crucial to emphasize that all profiles exhibit at least one virulence factor associated with invasion, adhesion, or infection, which initiates intramammary infection leading to bovine mastitis. The first major virulence profile includes *cfb*, *cyl*, *fbsA*, *fbsB*, *PI-2b*, and *bibA*, which were identified in 13 herds (*n* = 51). The second major profile consists of *cfb*, *cyl*, *fbsA*, *fbsB*, and *bibA*, identified in 13 herds (*n* = 45) (Table 3).

However, the presence of the virulence factors did not show a statistically significant association with the type of mastitis (clinical and subclinical).

### 3.2. Antimicrobial Susceptibility Profiles

Antimicrobial susceptibility was interpreted according to the CLSI VET 08 breakpoint zone diameter standard. *S. agalactiae* strains showed susceptibility to penicillin (95%), ampicillin (95%), amoxicillin-clavulanic acid (100%), cefotaxime (98%), ceftiofur (100%), erythromycin (99%), enrofloxacin (44%), sulfamethoxazole-trimethoprim (92%), tetracycline (68%), and vancomycin (96%). However, our collection exhibited an intermediate level of susceptibility to enrofloxacin (56%). All *S. agalactiae* strains were resistant to kanamycin (100%) and gentamicin (100%) (Figure 2).

We identified seven strains exhibiting multidrug resistance (MDR) profiles, including resistance to aminoglycosides, quinolones, tetracycline, beta-lactams, and glycopeptides (Table 4). At least four MDR profiles have been categorized. These four MDR profiles were found in only two virulence factor profiles (A and B).

The first MDR profile of *S. agalactiae* was from herd F, which was one of the eighteen strains (5.55%). This strain is resistant to five antimicrobials (penicillin, ampicillin, kanamycin, gentamicin, and sulfamethoxazole-trimethoprim). The second MDR *S. agalactiae* profile comprised four strains from three herds. The MDR strains included one from herd C (100%), one from herd D (11.11%), and two strains from herd K (50%). These strains are resistant to four antimicrobials (kanamycin, gentamicin, sulfamethoxazole-trimethoprim, and tetracycline). The third MDR profile of *S. agalactiae* was identified in herd D, which was one of the nine strains (11.11%). This strain is resistant to four antimicrobials (kanamycin, gentamicin, tetracycline, and vancomycin). The last MDR profile of *S. agalactiae* was also identified in herd D, which was one of the nine strains (11.11%). This strain is resistant to four antimicrobials (kanamycin, gentamicin, erythromycin, and vancomycin). It is important to highlight that herd D had three different MDR strains of *S. agalactiae*. We also identified two vancomycin-resistant strains of *S. agalactiae* with MDR profiles causing bovine mastitis in herd D.

## 4. Discussion

Bovine mastitis caused by *S. agalactiae* is a major veterinary and dairy farmer issues globally [1,13]. It is responsible for one of the main types of contagious mastitis in dairy farms [33]. The infection typically spreads from cow to cow during milking. The infected cow is the primary source of infection within the herd. In this study, we identified the virulence factor profiles and antimicrobial susceptibility of *S. agalactiae* causing bovine mastitis in Thailand’s intensive dairy production area.

This study investigated virulence factors related to pathogenicity, including the ability to adhere, invade host cells, and establish an infection in the mammary gland. It is important to highlight that we detected the virulence factors *bibA*, *fbsB*, *and cfb* in all *S. agalactiae* strains. The analysis of virulence factors revealed that the majority of the strains (96%) amplified the following virulence genes: *fbsA*, *fbsB*, *bibA*, *cfb*, and *cyl*, which play a role in the stages of intramammary infection caused by *S. agalactiae*, such as adhesion (*bibA*), invasion and colonization (*fbsA* and *fbsB*), and infection (*cfb* and *cyl*).

As for the adhesion, invasion, and colonization stages, our study found that all strains harbored *bibA* and *fbsB* genes. The *fbsA* gene was identified in most strains (98%). Our findings align with a previous study indicating that all bovine mastitis strains caused by *S. agalactiae* were positive for *bibA* [10,16] and *fbsB* [23]. The *fbsA* gene was identified in 92.6% of *S. agalactiae* isolates from dairy herds in Poland [23]. The *bibA*, *fbsA*, and *fbsB* genes are involved in the adhesion and invasion mechanisms of the pathogen in the mammary gland, which facilitates the attachment of *S. agalactiae* to mammary epithelial cells. The ability of bacterial cells to adhere is the most important mechanism for the successful colonization of mammary tissue. This mechanism enables bacteria to effectively neutralize the continuous flow of milk and establish a persistent presence in the bovine mammary gland [15]. This prolonged colonization may ultimately contribute to the development of chronic and subclinical mastitis [23]. Our study did not identify any associations between individual virulence factors and virulence profiles associated with types of mastitis (clinical and subclinical).

Thus, the *cyl* gene serves as a valuable marker for identifying and investigating the pathogenic potential of *S. agalactiae* in the context of infections. Several studies have confirmed the presence of this factor in all tested strains of both clinical and subclinical mastitis [15,34]. Furthermore, they are involved in the recruitment of various cytotoxic and pro-inflammatory cytokines, leading to the recruitment of neutrophils that cause tissue damage [21].

The study investigated hemolytic virulence by examining both the *cyl* gene and phenotypic hemolysis. Most *S. agalactiae* strains carried the *cyl* gene, at 97%, while 88% of the strains exhibited β-hemolysis on blood agar. These findings are consistent with a previous study indicating that 95.6% of *S. agalactiae* isolates were positive for the *cyl* gene, and 89.7% exhibited β-hemolysis [23]. The *cyl* gene encodes the cytolysin enzyme, a key factor in the hemolytic activity of group B streptococci [12,19]. The production of cytolysin leads to the lysis of red blood cells, resulting in the characteristic pattern of beta-hemolysis observed on blood agar. It is important to emphasize that all β-hemolytic strains in this study were consistently associated with the presence of the *cyl* gene. These findings support the presence of a strong correlation between gene expression and hemolysis. The remaining nine strains, despite harboring the *cyl* gene, did not exhibit β-hemolysis, which could be attributed to the lack of gene expression [19,23]. Certain Group B streptococci exhibit beta-hemolysis and harbor the entire *cyl* gene cluster, while other strains show no hemolysis and have mutations in the *cyl* locus [35,36].

The *cfb* gene encodes the CAMP factor, which promotes hemolysis. Notably, the *cfb* gene was detected in all strains in this study, which is consistent with other research [37,38]. Functionally, this gene forms pores in the host’s cell membrane and binds to glycosylphosphatidylinositol (GPI)-anchored proteins, leading to the subsequent impairment of the host’s immune function [12,19,39]. Previous studies suggest that the CAMP factor toxin plays a significant role in *S. agalactiae* infection [20,23].

Our study categorized a significant proportion of strains (51%) that displayed specific virulence gene profiles, including *cfb*, *cyl*, *fbsA*, *fbsB*, *PI-2b*, and *bibA* (Table 3). Furthermore, our analysis revealed that 45% of the strains exhibited a virulence gene pattern comprising *cfb*, *cyl*, *fbsA*, *fbsB*, and *bibA*, while concurrently lacking the *PI-2b* gene. The presence of *PI-1*, *PI-2a*, and *PI-2b* in *S. agalactiae* is known to facilitate processes such as adherence, invasion, and immune evasion [19]. Notably, research into *PI-2b* has revealed its strong capacity for adherence and biofilm formation in bovine strains. This evidence supports the capability of *S. agalactiae* to adhere to the host cell [15]. This study did not detect strains with *PI-1*, *PI-2a*, *bac*, and *bca*, consistent with other research on *S. agalactiae* isolated from clinical and subclinical bovine mastitis [11,22,34,40]. The *bac* and *bca* genes encode surface molecules that mediate adhesion to host cells. In humans, it has also been reported as a key factor in the invasion of human cervical epithelial cells [41].

In further studies, we aim to prioritize the selection of CPS serotype Ia exhibiting a virulent profile consisting of *cfb*, *fbsB*, and *bibA* genes. This selection is intended for rapid diagnosis and as a primary target in the development of candidate vaccines. These profiles have all of the virulence factors which are essential for establishing mastitis infection and match all strains we examined in specific regions of Thailand. Further research is required to fully assess the effectiveness of vaccine implementation in the future. This study did not show the relationship between the virulence profiles and MDR profiles. However, virulence profile B is the only profile showing vancomycin resistance.

Antimicrobial treatment is recommended for *S. agalactiae* causing bovine mastitis [42]. In our study, we assessed the susceptibility of the *S. agalactiae* strains to 12 different antimicrobials using the Kirby–Bauer disk diffusion susceptibility test. Our findings revealed varying degrees of susceptibility among the strains. Our *S. agalactiae* strains are predominantly susceptible to beta-lactam antibiotics (95–100%), including penicillin, ampicillin, cefotaxime, and ceftiofur. These findings align with a prior study in Brazil [43] and a study on of *S. agalactiae* in clinical bovine mastitis isolates from Denmark, which demonstrated full susceptibility to penicillin in all isolates [44]. In contrast, studies in China have reported significantly higher resistance rates to beta-lactam drugs in cases of bovine mastitis caused by *S. agalactiae* [45,46]. Important, a previous report has shown that the dissemination of these resistant pathogens in both humans and animals, along with potential environmental contamination caused by the utilization of manure as fertilizer, is associated with an increased global risk. Hence, a one health perspective is important [47,48]. These inconsistent findings may reflect the clinical use of antimicrobials across geographic regions. Nevertheless, penicillin and ampicillin remain the preferred treatment options for *S. agalactiae* which cause bovine mastitis, particularly in our study area.

In Thailand, antimicrobial drugs are extensively employed to address prevalent infectious diseases as part of common therapeutic practices. For example, the treatment of bacterial complications associated with foot-and-mouth disease [49], respiratory infections, such as bovine respiratory disease complex [50], reproductive tract disorders, and mastitis are among the diverse array of health challenges encountered in cattle. In the case of mastitis, the intramammary antimicrobial drugs which can be used include penicillin, ampicillin, cloxacillin, gentamicin, cephalothin, cefuroxime, and ceftiofur. All *S. agalactiae* strains were resistant to kanamycin and gentamicin. These findings align with a study from Northern Italy, wherein *S. agalactiae* isolates from humans and cattle exhibited resistance to aminoglycosides [51]. Furthermore, other studies have shown high resistance rate (93.3–100%) to kanamycin and gentamicin in clinical bovine mastitis isolates [24,52]. The aminoglycoside resistance mechanism in *S. agalactiae* remains poorly understood. Overall, aminoglycoside resistance encompasses various mechanisms, such as enzymatic modification, target site alteration through enzymatic action or chromosomal mutation, and efflux mechanisms [53]. Enzymatic modification is often found in plasmids harboring multiple resistance elements, including additional enzymatic modifications of aminoglycosides [54]. This is influenced by the selective pressure exerted by the excessive or improper use of these antimicrobials, which varies according to the geographic area of specific aminoglycoside use [55]. High rates of gentamicin resistance in Group B Streptococcus also result in the loss of a synergistic effect on susceptibility to beta-lactam treatment [56].

In this study, 56% of *S. agalactiae* strains exhibited intermediate susceptibility to enrofloxacin. A prolonged course of treatment and dosage adjustment is required. Other studies have reported evidence of quinolone resistance in China [46], Croatia [57], and Slovakia [58]. However, it is important to note that quinolone resistance is acquired through the accumulation of various resistance mechanisms within a single bacterial strain and can be transferred to others [59].

In this study, 32% of *S. agalactiae* strains are resistant to tetracycline. In a previous study, high resistance rates (95%) to tetracycline were reported in *S. agalactiae* strains isolated from bovine mastitis [60]. These findings are consistent with other research on antimicrobial susceptibility, showing that *S. agalactiae* is typically susceptible to a range of antimicrobials, with the exception of tetracycline [11,25]. Tetracycline resistance has been confirmed to frequently involve the presence of resistance genes among bovine isolates, which are disseminated through horizontal gene transfer within the same genus [61,62]. The resistance genes, which can be carried on mobile genetic elements such as plasmids or transposons, allow them to protect ribosomes from the inhibitory effects of tetracycline. The global decline in tetracycline efficacy is likely due to its past overuse, including in non-targeted treatments [52].

Interestingly, our study revealed that seven MDR *S. agalactiae* strains occurred in four dairy herds. In particular, herd D, we were able to detect three strains (33.33%; 3/9 strains) with different MDR profiles of *S. agalactiae*. The MDR profiles of *S. agalactiae* varied across antimicrobial groups. This is the first evidence of MDR *S. agalactiae* causing bovine mastitis in Thai dairy farms. The previous study on two human strains of *S. agalactiae* resistant to vancomycin was first reported in 2014 [63]. The high prevalence of vancomycin-resistant *S. agalactiae* causing bovine mastitis has been reported in small dairy farms in Turkey [64]. However, our study indicated that only 2% (2/100 strains) of *S. agalactiae* are resistant to vancomycin. These two strains are present among our seven MDR strains, which were identified in herd D.

The common antimicrobials used in Thai dairy farms include aminoglycoside (gentamicin and kanamycin), quinolones (enrofloxacin), tetracycline, sulfonamides (sulfamethoxazole-trimethoprim), and beta-lactams (penicillin and ampicillin). These agents are widely used for the treatment of infectious diseases in dairy cattle in Thailand, including bacterial complications from foot-and-mouth disease [49], respiratory infections such as bovine respiratory disease complex [50], gastrointestinal infections, blood parasite infections, reproductive tract infections, and mastitis. Our findings may raise awareness and concerns regarding the routine use of antimicrobials on dairy farms. This incidence of MDR *S. agalactiae* may pose a global public health risk [65]. This study provides new insights into MDR *S. agalactiae* causing bovine mastitis in Thailand. Furthermore, our study underscores the critical importance of ongoing surveillance of *S. agalactiae* to find epidemiological patterns and antimicrobial susceptibility, in conjunction with global research on the emerging trends of MDR GBS isolates.

Extensive use of antimicrobials on dairy farms likely contributes to their high prevalence of AMR [66]. Over 50% of antimicrobial agents used in treating bovine mastitis led to excessive and unnecessary economic losses for dairy farms [67]. Furthermore, this treatment practice carries the risk of fostering the emergence of bacterial resistance and antimicrobial residues in milk [68]. They pose a threat not only to human health but also to the environment as pollution [45]. The incidence of MDR in this study encourages farmers, veterinarians, consumers, and policymakers to prioritize the rational use of antimicrobials to minimize the risk of MDR emergence in animals and humans.

It is essential to identify, characterize, and monitor AMR in bacterial pathogens using classical microbiological and molecular approaches. The comparison between human and bovine-derived strains [2] should be more interested in order to investigate the linkage transmission profiles of these agents [2,69,70,71]. Further genotypic study of the AMR resistance gene enhances our understanding of the mechanics and epidemiology of AMR. It also serves as a guide for treatment decisions and infection control strategies. Accountability for the use of antimicrobials in animals and adhering to proper guidelines and regulations to minimize unnecessary antimicrobial use are urgently required. Improving milking hygiene and sanitation practices may enhance efforts to the spread of this pathogen and decrease antimicrobial usage.

## 5. Conclusions

The study on the characteristics of Thai strains of *S. agalactiae* causing bovine mastitis updates our understanding of this contagious mastitis pathogen. We found that all strains have at least three virulence factors (*bibA*, *fbsB*, and *cfb*) which are implicated in adhesion, invasion, and the intramammary infection stages during mastitis establishment. These findings aid in the advancement of bovine mastitis prevention and control strategies, encompassing vaccine development prospects. The beta-lactam antimicrobial class is effectively used in treating cows infected with *S. agalactiae*. MDR strains of *S. agalactiae* associated with bovine mastitis have been identified in our region. Antimicrobial susceptibility profile monitoring should be conducted regularly. The rise in AMR among bovine strains of *S. agalactiae* calls for significant concern and comprehensive integrated measures regarding the justified use of antimicrobials on Thai dairy farms.

## Figures and Tables

**Figure 1 animals-14-00447-f001:**
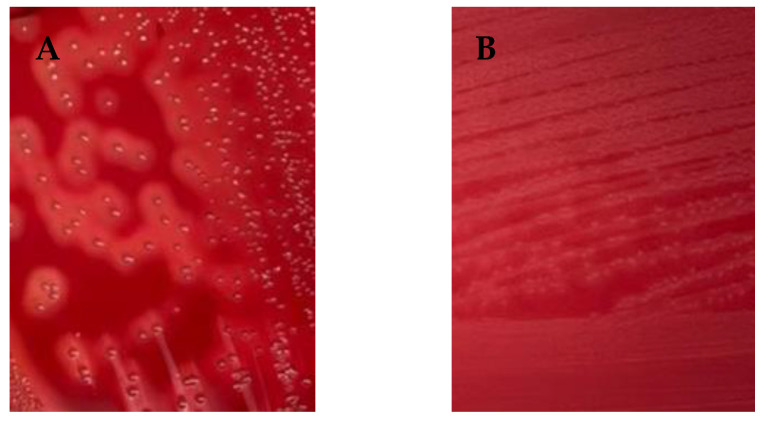
Hemolytic phenotype activity of *S. agalactiae* field strains. (**A**) shows β hemolysis (*n* = 88), (**B**) shows α hemolysis (*n* = 12).

**Figure 2 animals-14-00447-f002:**
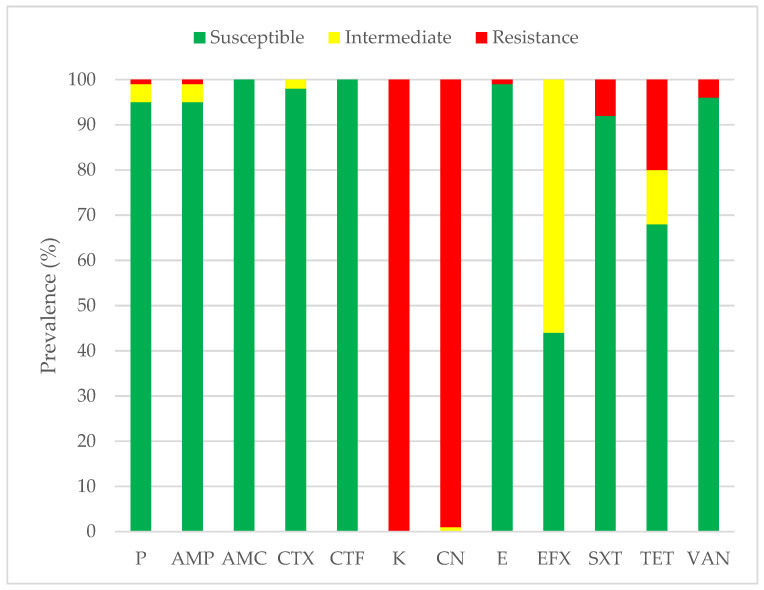
Antimicrobial susceptibility profile of *S. agalactiae* (*n* = 100). Abbreviations: P: penicillin; AMP: ampicillin; AMC: amoxicillin; CTX: cefotaxime; CTF: ceftiofur; K: kanamycin; CN: gentamicin; E: erythromycin; EFX: enrofloxacin; SXT: sulfamethoxazole-trimethoprim; TET: tetracycline and VAN: vancomycin.

**Table 1 animals-14-00447-t001:** Characteristics of 100 *S. agalactiae* strains associated with mastitis type.

Herds	Province	Year	Herd Size ^1^	Number of Strains	Number of Cases	Number of Clinical Cases	Number of Subclinical Cases
Total strains/Cases				100	58	21	37
A	Saraburi	2018–2019	L	31	22	12	10
B	Nakorn-Ratsima	2016	M	1	1	0	1
C	Saraburi	2017	S	1	1	0	1
D	Nakorn-Ratsima	2016–2017	M	9	4	0	4
E	Saraburi	2018	S	1	1	1	0
F	Nakorn-Ratsima	2016–2017	M	18	6	2	4
G	Nakorn-Ratsima	2017	S	3	3	0	3
H	Saraburi	2017	M	3	2	0	2
I	Saraburi	2017	S	8	3	0	3
J	Saraburi	2016	M	15	8	2	6
K	Saraburi	2017	M	4	2	0	2
L	Ratburi	2019	L	5	4	4	0
M	Nakorn-Ratsima	2019	M	1	1	0	1

^1^ Herd size—approximate number of lactating cows per herd during the study period; L, large herd ≥100 dairy cows; M, medium herd = 21–100 dairy cows; S, small herd ≤ 20 dairy cows; herd size is classified according to Koonawootrittriron and Elzo [28].

**Table 2 animals-14-00447-t002:** The list of primers used to determine virulence factors; data from [15,29,30].

Function	Gene	Sequence Forward (5′ to 3′)	Tm (°C)	Sequence Reverse (5′ to 3′)	Tm (°C)
Adhesion	*bibA*	AATCGAAAACAACGTTGGAAAG	52.3	AAACCAGGCTTCATCAGTCATT	54.7
	*bca*	CTACAATTCCAGGGAGTGCA	54.5	ACTTTCTTCCGTCCACTTAG	51.8
	*bac*	AAGCAACTAGAAGAGGAAGC	52.1	TTCTGCTCTGGTGTTTTAGG	52.3
Invasion	*fbsA*	GTCACCTTGACTAGAGTGATTAT	51.6	CCAAGTAGGTCAACTTATAGGGA	53.4
	*fbsB*	TCTGTCCAACAGCCGGCTCC	62.3	TTCCGCAGTTGTTACACCGGC	60.5
	*PI-1*	AACAATAGTGGCGGGGTCAACTG	59.6	TTTCGCTGGGCGTTCTTGTGAC	60.4
	*PI-2a*	CACGTGTCGCATCTTTTTGGTTGC	59.6	AACACTTGCTCCAGCAGGATTTGC	60.4
	*PI-2b*	AGGAGATGGAGCCACTGATACGAC	59.9	ACGACGACGAGCAACAAGCAC	60.4
Infection and Tissue damage	*cfb*	AAGCGTGTATTCCAGATTTCC	52.9	AGACTTCATTGCGTGCCAAC	56.2
	*cyl*	ACGGCTTGAACGACGTGACTAT	58.6	CACCAATTGGCAGAGCCT	55.6

**Table 3 animals-14-00447-t003:** The presence of virulence factor profiles in *Streptococcus agalactiae* strains causing bovine mastitis.

Profiles	Virulence Factors										Number of Strains	Herds
	Adherence			Invasion and Colonize					Infection			
	*bibA*	*bca*	*bac*	*fbsA*	*fbsB*	*PI-1*	*PI-2a*	*PI-2b*	*cfb*	*cyl*		
ATCC 12400CPS type Ia	+	−	−	+	+	−	−	+	+	+	Reference	
ATCC 13813CPS type II	+	−	−	+	+	−	+	+	+	+	Reference	
ATCC 31475CPS type III	+	+	+	+	+	+	+	−	+	+	Reference	
A	+	−	−	+	+	−	−	+	+	+	51	A–M
B	+	−	−	+	+	−	−	−	+	+	45	A–M
C	+	−	−	+	+	−	−	+	+	−	2	A and F
D	+	−	−	−	+	−	−	−	+	+	1	J
E	+	−	−	−	+	−	−	−	+	−	1	A
Total	100	0	0	98	100	0	0	53	100	97	100	

Noted: The presence of virulence factors was categorized in five profiles. Category A: presence of *bibA*, *fbsA*, *fbsB*, *PI-2b*, *cfb*, and *cyl*. Category B: presence of *bibA*, *fbsA*, *fbsB*, *cfb*, and *cyl.* Category C: presence of *bibA*, *fbsA*, *fbsB*, *PI-2b*, and *cfb.* Category D: presence of *bibA*, *fbsB*, *cfb*, and *cyl*. Category E: presence of *bibA*, *fbsB*, and *cfb.*

**Table 4 animals-14-00447-t004:** Multidrug-resistant *S. agalactiae* profiles.

Multidrug Resistance Profile	Virulence Profiles	Number of Strains	Herds (*n*)
Penicillin, Ampicillin, Kanamycin, Gentamicin, Sulfamethoxazole-Trimethoprim	A	1	F (1)
Kanamycin, Gentamicin, Sulfamethoxazole-Trimethoprim, Tetracycline	A (3) and B (1)	4	C (1), D (1), K (2)
Kanamycin, gentamicin, tetracycline, vancomycin	B	1	D (1)
Kanamycin, gentamicin, erythromycin, vancomycin	B	1	D (1)

Noted: Virulence profiles as described in Table 3.

## Data Availability

The data presented in this study are available on reasonable request from the corresponding author.

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
