# Peer review of "Characterization of Virulence Factors and Antimicrobial Susceptibility of Streptococcus agalactiae Associated with Bovine Mastitis Cases in Thailand"

_animals, 2024, doi:10.3390/ani14030447_

Round 1

Reviewer 1 Report

Comments and Suggestions for Authors

The presented manuscript tries to describe the molecular carachteristics and the phenotypic antimicrobial resistance patterns of several Streptococcus agalactiae isolates derived from bovine mastitis clinical and subclinical cases.

The manuscript is well written, with good description of methodology and clear results presentation.

The subject analysed here is of prime importance in the dairy production industry since mastitis takes a big toll in economic losses due to the presence of the pathology in producing animals. Even more important is the fact that antibiotic treatment is more and more considered a last resort due to the incresing rates of antimicrobial resistance worldwide.

The manuscript needs only minor modifications that i will list below:

Line 61-62: You should specify the name of the "critical virulence factor" in order to make the phrase more clear and devoid of any possible misinterpretation.

Paragraph 2.5: please state if the PCR performed is singleplex or multiplex. It would also be advisable to state the Tm of each primer used.

Table 3: please highlight in grey the + related to PI-2b and profile C.

Author Response

Please see the attachment. The attached file is composed of the revised manuscript and our respond to the reviewer comments.

Reviewer 2 Report

Comments and Suggestions for Authors

Thank you for your work. See my comment and correction in the attached version. Please use Adobe to correctly see them.

Comments on the Quality of English Language

Some correction should be amended prior to publication

Author Response

(The authors gave the same response as above.)

Reviewer 3 Report

Comments and Suggestions for Authors

Ajariyakhajorn et al have investigated the Antimicrobial Susceptibility and Virulence Factors of Streptococcus agalactiae isolated from bovine mastitis cases in Thailand. The finding from this study provides crucial information and serves as a guide for treatment decisions, and protective and control measures for bovine mastitis in Thailand.

This study is fascinating, relevant, and well-written. However, some minor issues need to be corrected before considering this manuscript for publication.

 Comments to Authors

1.      1- There is no mentioned the antibiotic disk concentration for all tested antibiotic in this study. Therefore, authors must include them in the method section.

2- The CLSI document cited is a human document. It must use a animal document since the bacteria isolated from animals.

Author Response

(The authors gave the same response as above.)

Reviewer 4 Report

Comments and Suggestions for Authors

The manuscript is well written and the results is well presented. I cannot recommend that this article be published in present format because the because of some basic issues that need to be corrected as below:

1-     In the first line of simple summary the name of organism “S. agalactiae” should be in full form “Streptococcus agalactiae”.

2-     Line-21, “we identified” should be changed to “we detected”

3-     Line-26, the genes name should be written in italic.

4-     Please specify the tests used to detect and identify bacterial isolates in the method section.

5-     Please specify the medium used for the stock of bacteria in -80 freezer.

6-     Please specify the concentration of antibiotics in the disks , for example Vancomycin (30µg).

7-     Line-174, the name of microorganism (S. pneumoniae) should be in full form (Streptococcus pneumoniae)

8-     Statistical analysis should be added to the methods section.

9-     Lines 183-184, should be transfer to the method section.

10-  Line 223-227, should be deleted. These lines are repeated in the methods section and should be deleted.  

11-  The relationship between the presence of virulence factors and antibiotic resistance should be addressed in the “Results” section and discussed in the “Discussion” section.

12-  Please report the co-occurrence profile of virulence genes in a Table (as Table- 4 designed for antibiotics resistance profiles).

Comments on the Quality of English Language

Minor editing of English language required

Author Response

(The authors gave the same response as above.)

Round 2

Reviewer 2 Report

Comments and Suggestions for Authors

Thank you for the improvements added in the revised verison of your manuscript

Reviewer 4 Report

Comments and Suggestions for Authors

The authors have considered the suggestions and comments presented in the previous review in the manuscript.  

Comments on the Quality of English Language

 Minor editing of English language required.